# An Absorbing Markov Chain Model to Predict Dairy Cow Calving Time

**DOI:** 10.3390/s21196490

**Published:** 2021-09-28

**Authors:** Swe Zar Maw, Thi Thi Zin, Pyke Tin, Ikuo Kobayashi, Yoichiro Horii

**Affiliations:** 1Interdisciplinary Graduate School of Agriculture and Engineering, University of Miyazaki, Miyazaki 889-2192, Japan; z3t1802@student.miyazaki-u.ac.jp; 2Graduate School of Engineering, University of Miyazaki, Miyazaki 889-2192, Japan; pyketin11@gmail.com; 3Field Science Center, Faculty of Agriculture, University of Miyazaki, Miyazaki 889-2192, Japan; ikuokob@cc.miyazaki-u.ac.jp; 4Center of Animal Disease Control, University of Miyazaki, Miyazaki 889-2192, Japan; horii@cc.miyazaki-u.ac.jp

**Keywords:** absorbing Markov chain, cow behavior analysis, prediction of calving time

## Abstract

Abnormal behavioral changes in the regular daily mobility routine of a pregnant dairy cow can be an indicator or early sign to recognize when a calving event is imminent. Image processing technology and statistical approaches can be effectively used to achieve a more accurate result in predicting the time of calving. We hypothesize that data collected using a 360-degree camera to monitor cows before and during calving can be used to establish the daily activities of individual pregnant cows and to detect changes in their routine. In this study, we develop an augmented Markov chain model to predict calving time and better understand associated behavior. The objective of this study is to determine the feasibility of this calving time prediction system by adapting a simple Markov model for use on a typical dairy cow dataset. This augmented absorbing Markov chain model is based on a behavior embedded transient Markov chain model for characterizing cow behavior patterns during the 48 h before calving and to predict the expected time of calving. In developing the model, we started with an embedded four-state Markov chain model, and then augmented that model by adding calving as both a transient state, and an absorbing state. Then, using this model, we derive (1) the probability of calving at 2 h intervals after a reference point, and (2) the expected time of calving, using their motions between the different transient states. Finally, we present some experimental results for the performance of this model on the dairy farm compared with other machine learning techniques, showing that the proposed method is promising.

## 1. Introduction

Even though calving is a normal physiological process, it is important to manage not only for the sake of the animals’ welfare, but also for ensuring economic growth in the dairy industry [1,2]. Accurately predicting calving time helps overcome the difficulties of parturition, providing human assistance when needed, and reducing calf mortality. The problem of observing behavioral changes for predicting calving time has been widely studied [3,4]. Solutions are typically sensor-based systems that require the use of wearable or non-wearable sensors to monitor daily behavior and provide responses when calving is imminent. Wearing sensors all the time might cause great discomfort for pregnant cows, as well as risk damaging the sensors themselves while cows move around in the barn [5].

In this study, we focused on a camera-based system because it can support a smart, adjustable, time and money saving way to monitor what happens in the calving barn, and the condition of cows. Best of all, the system allows tracking everything in real time right on our PC, smartphone, or tablet. However, the system has advantages and disadvantages. Similarities between the background and the cow’s body color complicates the detection of cows, and clearly identifying each individual cow is still limited in using surveillance cameras over the long-term. For this reason, we will use advanced image processing technology to solve these problems in the future. Figure 1 illustrates our research work.

The dairy farming industry has benefited greatly because of advances in Information and Communications Technology (ICT), as well as in Artificial Intelligence (AI) and the Internet of Things (IoT). Smart dairy farms are no longer out of our reach. The intelligent and efficient monitoring of individual cows will be a necessary part of making dairy farms smart. Among many other issues, caring for pregnant cows is crucial to dairy farm management, especially when calving occurs. Insufficient monitoring at the time of parturition can be extremely detrimental [6,7,8]. It can prolong the process of giving birth and increase the risk of both stillbirth and calving difficulties, causing impaired reproductive performance and increased calving-to-conception intervals.

In this research, we concentrated on monitoring daily routines for the purpose of detecting when behavioral changes increase in frequency, signaling the approach of parturition, and allowing a prediction of the exact calving time. We describe how we approached this goal by designing a system that features observations of video data collected using a 360-degree camera on an hourly basis for 72 h before the start of calving. For this reason, we propose a new method based on image processing techniques and Markov chain model to predict the time at which cow calving will occur. We also compare our proposed method with three other machine learning techniques: K-nearest Neighbors (KNN), Naïve Bayes (NB), and Support Vector Machine (SVM).

Specifically, we analyze the behavior of pregnant cows in maternity barns by embedding this behavior into a Markov chain, thus predicting the time of calving. Model performance was evaluated on video data, which were collected from 25 dairy cows at Oita Prefecture in Japan, to verify that our proposed method has potential as a method for predicting calving time. From these videos taken in the maternity barn, human observers used a reversible counter system in 5 min increments to record the number of changes in lying posture, the number of transitions from lying to standing, the number of changes in standing posture, and the number of transitions from standing to lying. All these data for statistical analysis of behavior were collected for three days before the predicted calving date.

Results of this analysis showed that the proposed methodology could be applied and achieved plausible results. Our analysis indicates that investigating behavioral activity peaks in these data will be useful in improving the prediction process. However, additional avenues should be explored in pursuing research on calving time prediction. As an alternative, the application of Markov modeling [9] to predict calving time is an appealing methodology. The most frequent applications of augmented absorbing Markov chain modeling are in predicting future stock exchange trends [10], predicting web user behavior [11], and forecasting educational attainment rates [12]. Our current concern lies in using an absorbing Markov chain to develop a prediction model from observations of the behavioral changes of the dairy cows. Specifically, we propose in this paper the use of such a model to predict when calving will occur. By comparing the results of predicting calving events using our proposed method with the results with other machine learning techniques, we see that our proposed method more accurately predicts calving events.

The organization of this paper is as follows. Section 2 follows with a description of the proposed method for calving time prediction. We show some experimental results and discussion in Section 3 and Section 4. Finally, we conclude our approach in Section 5 with some suggestions and discussion of possible future research.

## 2. Materials and Methods

### 2.1. Data Collection and Preparation

The animal experimentation protocol of ethical statement and approval were granted for this study, animals were neither enforced nor uncomfortably restricted during the study period. The video data of monitoring calving process used for analysis in this study were collected by an installed camera without disturbing natural parturient behavior of animals and routine management of the farm. 

The experimental design is established on a large dairy farm situated in Oita Prefecture, Japan. Three primiparous and 22 multiparous pregnant dairy cows were housed in roofed cowsheds. Four or five pregnant cows were together housed in a calving pen. which was 7 × 7 m^2^ with sawdust flooring for when calving event was close to occurring. The cows were fed with Total Mixed Ration (TMR) twice daily for their maintenance and pregnancy, as calculated based on each cow’s body weight and the expected average milk yield (35 kg/day) after giving birth. They were also provided ad libitum access to clean water and mineral supplements.

Experimented cows were continuously monitored using a 360-degree GV-FER5700 camera (Geo Vision Inc., Taiwan, China) (2560 × 2048 pixels, recording at 30 frames per second), which was set up 3 m above the pregnant cows located in the maternity barns. This camera can capture images within a 360-degree field of view in the horizontal plane. Using this camera, the positions and states of cows appear in different parts of the 360-degree view. All cows are clearly visible from overhead, allowing a determination of the condition of each cow. The video sequences for pregnant cows are continuously collected until calving occurs.

In this section, we describe how an augmented Markov chain model can be employed to predict a calving event. To do so, we firstly prepare the dataset from video sequences taken during the three days before calving. After collecting the video sequences, the target cow regions are manually extracted by Visual Geometry Group (VGG) annotator [13] to remove the background, and to obtain the cow contour regions by using image processing techniques.

We used an approach based on statistical analysis to predict calving time. Cow behavioral activities were recorded by human observers performing a direct visual observation of each individual cow in the calving barn. The four types of conditions include two postures and two transitions. They are defined as follows.
L (Posture): lying in the calving barn;LS (Transition): rising from a lying state to a standing state;S (Posture): standing on all four legs;SL (Transition): changing from a standing state to a lying state.

Images are labeled to count the number of pairs from one state to another in making a co-occurrence matrix for the 72 h before calving. Although cows may assume many other states such as eating and drinking, all other activities are assumed to be subsets of the above-mentioned activities. Because of this, our video recording only concerns a sequence of the four activities for each individual cow, continuously monitored until calving occurs. Figure 2 illustrates a sample of the four posture conditions. In Figure 3, the system architecture of our proposed method is represented.

Generally, absorbing Markov Chain is used to investigate behaviors of any state which eventually enter an end state or absorbing state. In our case, the absorbing state is the event at which calving occurs. In theory of Markov Chain, we can compute the time entering an absorbing state and the probability of absorbing. So, we thought that it would be tractable to apply the absorbing Markov Chain Model for the calving time of a pregnant dairy cow. Although there have been many applications in queuing and dam theory [14], we have not seen any application of absorbing Markov Chain for the prediction of dairy cow calving process. So, at this stage we have not made comparison with previous methods in this aspect.

Moreover, the nature of absorbing state from the absorbing Markov chain is similar with the nature of calving event in prediction calving time model in dairy cows. From the concept of absorbing Markov chain theory, the absorbing state is end-state, and the calving state is also end-state in the prediction of calving time model.

### 2.2. Creation of Co-Occurrence Matrix and Markov Chain Model

The co-occurrence and probability matrices of the Markov chain model are created using the state sequence described in Figure 2. In order to do so, we first define the number of co-occurrences of state pairs. Let $c\left(s_{i},\;s_{j}\right)$ be the number of pairs of states $\left(s_{i},\;s_{j}\right)$ for i,j=1,2,3,4,5, and $s_{1}=L,\;s_{2}=\mathrm{LS},\;s_{3}=S,\;s_{4}=\mathrm{SL},\;s_{5}=\mathrm{Calve}$. We can then have the corresponding co-occurrence matrix **C**, as shown below.
$$C=\left(c\left(s_{i},s_{j}\right)\right)=\left[\begin{array}{ccccc} c\left(s_{1},s_{1}\right) & c\left(s_{1},s_{2}\right) & c\left(s_{1}{,s}_{3}\right) & c\left(s_{1},s_{4}\right) & c\left(s_{1},s_{5}\right)\\ c\left(s_{2},s_{1}\right) & c\left(s_{2},s_{2}\right) & c\left(s_{2},s_{3}\right) & c\left(s_{2},s_{4}\right) & c\left(s_{2},s_{5}\right)\\ c\left(s_{3},s_{1}\right) & c\left(s_{3},s_{2}\right) & c\left(s_{3},s_{3}\right) & c\left(s_{3},s_{4}\right) & c\left(s_{3},s_{5}\right)\\ c\left(s_{4},s_{1}\right) & c\left(s_{4},s_{2}\right) & c\left(s_{4},s_{3}\right) & c\left(s_{4},s_{4}\right) & c\left(s_{4},s_{5}\right)\\ c\left(s_{5},s_{1}\right) & c\left(s_{5},s_{2}\right) & c\left(s_{5},s_{3}\right) & c\left(s_{5},s_{4}\right) & c\left(s_{5},s_{5}\right) \end{array}\right].$$

The above co-occurrence matrix, **C** can be written as (1):(1)$$C=\left(c_{ij}\right),$$
where, $c_{ij}=\#\left\{\left(i,j\right)|i,j\in S=\left\{1,2,3,4,5\right\}\right\}$.

We can then deduce the one step transition probabilities, $p_{ij}$ by defining (2):(2)$$p_{ij}=c_{ij}/\sum_{j=1}^{5}c_{ij},$$
which represents the one step transition probability of going from state *i* to state *j* in a Markov Chain. We then have the transition probability matrix, $P=\left(p_{ij}\right)$.

The sum of the row probabilities is equal to one, since each health state is independent of the others, and an animal must move to one of the five states. The diagonal represents the probability of staying in the same state. A state *k* in a Markov chain is defined as *absorbing* if $p_{kk}=1$, in other words all $p_{kj}=0$ for j≠k. In this study, the absorbing state is the calving state. When calving occurs, further investigation stops because we have achieved the objective of predicting the calving time. Thus, five of the states are considered transient states in the Markov chain, since each one can independently transition to another state. The four transient states are lying (L), transition from lying to standing (LS), standing (S), transition from standing to lying (SL). Calving is the absorbing state. Figure 4 describes this five-state absorbing Markov chain used to predict calving time in dairy cows.

### 2.3. Description of Calving Event

As an attractive feature of Markov models, they can describe the course of calving events over time. This is especially attractive for modeling calving since a cow’s state of behavior while calving influences the prediction of calving time. The transition probability matrix **P** summarizes the probabilities of cow activities and can be used to describe the probability of calving for an individual cow with a known activity state. The elements of the probability matrix $p_{ij}$ in the $i^{th}$ row and $j^{th}$ column is denoted by $p_{ij}\left(t\right)$, which represents the probability of a transition from state *i* to state *j* during *t* periods or *t* steps, where *t* measures in minutes. For an *m* state Markov model, the probability of the system visiting state *k* at time *t* can be denoted as $p_{k}\left(t\right)$. Therefore, for all *m* states, these probabilities can be expressed as a row vector, $p\left(t\right)=\left[p_{1}\left(t\right),\;p_{2}\left(t\right),\;\cdots ,\;p_{m}\left(t\right)\right]$. By using total probability, this equation can be written as (3) or (4):(3)$$p_{k}\left(t\right)=\sum_{j=1}^{m}p_{kj}p_{j}\left(t-1\right)\;\mathrm{for}\;k=1,\;2\;,\;\cdots ,\;m,$$
(4)$$p\left(t\right)=p\left(t-1\right)P^{1}=p\left(t-2\right)P^{2}=\cdots =p\left(0\right)P^{t},$$

Transient analysis may cause convergence of the probability distribution vector when *t* becomes very large. That is, as the number of steps *t* increases, the probability vector approaches a limiting value which is called the stationary distribution of the Markov chain. This stationary distribution, or as it is also called, steady state distribution, is denoted by $\pi =\left[\pi_{1},\;\pi_{2},\;\cdots ,\;\pi_{m}\right]$ and satisfies (5).
(5)π=πP,

Our case features five states: L, LS, S, SL, and Calve. Therefore, we get the following probabilities. $p_{i}\left(t\right)$ is the probability that the cow is in one of the states after a period of *t* from the start. In these ways, the behavior of dairy cows can be analyzed during periods when calving is imminent. These results will be described in the next section on experimental work.

### 2.4. Calving Time Prediction Procedure

This section presents the method of predicting calving time using the Markov chain model discussed in the previous section. This involves adding or augmenting the calving event as an absorbing state in our four-state Markov chain model. Since the problem is to predict the time of calving, the exercise is completed when the event occurs. Because of this, the calving state is considered absorbing. This means that any of the other four states can transition directly to the calving state, but once there, no additional transitions will occur. The probability of transitioning from absorbing state to absorbing state is one; and the probability of transitioning from absorbing state to any other state is zero. The four-state Markov chain model is transformed as described in (3) into an augmented Markov chain model of five states by adding an absorbing state (calving state).

#### Fundamental Matrix Solution: Absorbing Markov Chain

The matrix solution provides an exact solution for the time spent in each state, conditional on the entry state in which an individual enters the model. Such a matrix solution is only viable in time with homogeneous Markov chains with *r* absorbing states and *m* transient states. The transition probability matrix of a chain that contains an absorbing state is defined as the separation of a probability transition matrix **A** using canonical form.
(6)$$A=\left[\begin{array}{cc} Q & R\\ O & I \end{array}\right],$$
where, **I** is an *r*-by-*r* identity matrix, **O** is an *r*-by-*m* zero matrix, **R** is a nonzero *m*-by-*r* matrix and **Q** is an *m*-by-*m* matrix.

In the proposed augmented Markov model, **Q** is the matrix that contains transition probabilities between transient states, R is column vector of the calving state probabilities, **O** is the row vector of zero matrix, and **I** in [1] is 1 × 1 matrix. The iterated multiplication of the augmented matrix **A** yields as follows.
(7)$$A^{2}=\left[\begin{array}{cc} Q & R\\ O & I \end{array}\right]\times \left[\begin{array}{cc} Q & R\\ O & I \end{array}\right]=\left[\begin{array}{cc} Q^{2} & QR+R\\ O & I \end{array}\right],$$
(8)$$A^{3}=\left[\begin{array}{cc} Q^{2} & QR+R\\ O & I \end{array}\right]\times \left[\begin{array}{cc} Q & R\\ O & I \end{array}\right]=\left[\begin{array}{cc} Q^{3} & Q^{2}R+QR+R\\ O & I \end{array}\right],$$

Hence, by induction, we obtain the following:(9)$$A^{t}=\left[\begin{array}{cc} Q^{t} & Q^{t-1}R+Q^{t-2}R+\cdots +R\\ O & I \end{array}\right]\times \left[\begin{array}{cc} Q & R\\ O & I \end{array}\right]=\left[\begin{array}{cc} Q^{t} & \left(Q^{t-1}+\cdots +I\right)+R\\ O & I \end{array}\right].$$

However, when *t* tends to infinity the transient state matrix, $Q^{t}$ will tend to O (zero matrix). We then have from (6) that,
(10)$$A^{\infty }=\left[\begin{array}{cc} O & NR\\ O & I \end{array}\right],$$
where, $N=I+Q^{1}+Q^{2}+Q^{3}+\cdots ={\left(I-Q\right)}^{-1}$.

The matrix $N={\left(I-Q\right)}^{-1}$ is called the fundamental matrix for the augmented Markov chain model. Let N(i, j) be the element in row *i* and column *j*. Then, we can interpret the summation of N(i, j) over *j* as the expected number of periods until *absorbing* (calving). Therefore, the expected time until the *absorbing* (calving state) occurs is shown as $\sum_{j}\sum_{i}N\left(i,\;j\right)$. The probability of *absorbing* or calving at the expected time is p(0)×N×R.

## 3. Results

Data were collected on 25 dairy cows; 21 Holstein Black and White cows and 4 Brown Swiss cows were moved into the maternity barns from beginning 3 days before the expected calving date. We divided the 25 cows into 2 groups based on the primiparous and multiparous pregnant cows, as shown in Table 1 and Table 2. They calved at ages between 21 and 26 months and the calving period was between November and December in 2017. None of the 25 test cows presented with dystocia, and assistance for newborns calves was provided, as necessary. All newborn calves were single birth. Behavior analysis data were not collected after calving. Individual cows were continuously monitored until the calving event occurred using a 360-degree camera above the pregnant cows in the maternity barns.

Specifically, the four relevant activities were lying, transitions from lying to standing, standing, and transitions from standing to lying. From the collected videos, a sequence of the four activities is extracted for each cow as shown in the previous section. The co-occurrence matrix is constructed from the activity sequence for each individual cow. Sample co-occurrence matrices **C** are described below for Identity Document 2 (ID 2), Identity Document 11 (ID 11), and Identity Document (ID 27).
$$C_{\mathrm{ID}2}=\left[\begin{array}{ccccc} 1360 & 67 & 0 & 0 & 1\\ 0 & 0 & 65 & 4 & 1\\ 0 & 0 & 2686 & 68 & 1\\ 66 & 3 & 0 & 0 & 1\\ 0 & 0 & 0 & 0 & 1 \end{array}\right]$$
$$C_{\mathrm{ID}11}=\left[\begin{array}{ccccc} 1324 & 38 & 0 & 0 & 1\\ 0 & 0 & 38 & 4 & 1\\ 0 & 0 & 2834 & 38 & 1\\ 38 & 5 & 0 & 0 & 1\\ 0 & 0 & 0 & 0 & 1 \end{array}\right]$$
$$C_{\mathrm{ID}27}=\left[\begin{array}{ccccc} 1229 & 29 & 0 & 0 & 1\\ 0 & 0 & 29 & 2 & 1\\ 0 & 0 & 2972 & 28 & 1\\ 28 & 2 & 0 & 0 & 1\\ 0 & 0 & 0 & 0 & 1 \end{array}\right]\;$$

By row normalization, we obtain the Markov chain probability matrices **P** for ID 2, ID 11, and ID 27 as follows.
$$P_{\mathrm{ID}2}=\left[\begin{array}{ccccc} 0.952 & 0.047 & 0 & 0 & 0.001\\ 0 & 0 & 0.890 & 0.096 & 0.014\\ 0 & 0 & 0.976 & 0.024 & 0\\ 0.957 & 0.029 & 0 & 0 & 0.014\\ 0 & 0 & 0 & 0 & 1 \end{array}\right]$$
$$P_{\mathrm{ID}11}=\left[\begin{array}{ccccc} 0.971 & 0.028 & 0 & 0 & 0.001\\ 0 & 0 & 0.884 & 0.093 & 0.023\\ 0 & 0 & 0.986 & 0.013 & 0\\ 0.864 & 0.114 & 0 & 0 & 0.023\\ 0 & 0 & 0 & 0 & 1 \end{array}\right]$$
$$P_{\mathrm{ID}27}=\left[\begin{array}{ccccc} 0.976 & 0.023 & 0 & 0 & 0.001\\ 0 & 0 & 0.906 & 0.063 & 0.031\\ 0 & 0 & 0.990 & 0.009 & 0\\ 0.903 & 0.065 & 0 & 0 & 0.032\\ 0 & 0 & 0 & 0 & 1 \end{array}\right]$$

### 3.1. Calving Event as an Absorbing State of Markov Chain Model Implementation

The transition probability matrix defined as a Markov chain probability matrix is regular. The calving event is added as an absorbing state of the Markov chain as described above. The **Q** matrix for cow ID 2 is as follows.
$$Q_{\mathrm{ID}2}=\left[\begin{array}{cccc} 0.952 & 0.047 & 0 & 0\\ 0 & 0 & 0.890 & 0.096\\ 0 & 0 & 0.976 & 0.024\\ 0.957 & 0.029 & 0 & 0 \end{array}\right]\;$$

$R_{\mathrm{ID}2}={\left[\begin{array}{cccc} 0.001 & 0.014 & 0 & 0.014 \end{array}\right]}^{T}$;

$O_{\mathrm{ID}2}=\left[\begin{array}{cccc} 0 & 0 & 0 & 0 \end{array}\right]$; and $I_{\mathrm{ID}2}=\left[1\right]$;

$N_{\mathrm{ID}2}\;\mathrm{is}\;{\left(I_{\mathrm{ID}2}-Q_{\mathrm{ID}2}\right)}^{-1}$ which is given by:$$N_{\mathrm{ID}2}=\left[\begin{array}{ccccc} 372.492 & 17.969 & 657.388 & 17.487 & 1065.335\\ 356.900 & 18.245 & 667.502 & 17.756 & 1060.402\\ 361.332 & 17.459 & 679.840 & 17.977 & 1076.608\\ 366.725 & 17.720 & 648.286 & 18.245 & 1050.975 \end{array}\right]\;$$

Thus, from the theory developed in our method proposed in Section 3, the sum of all entries gives the expected time at which the calving event occurs. We obtain the predicted calving time as, $\sum_{j}\sum_{i}N\left(i,\;j\right)$ = 4253.320 min = 70.889 h from the beginning. The actual calving time is 72 h from the beginning. Therefore, our proposed method provides an accurate prediction. The probability of calving is expressed in the previous section. Thus, the probability of calving is certainly almost 1.

Similarly, we have derived the most useful statistics such as the co-occurrence matrices and their corresponding probabilities for all cows in this study. By adding the concept of an absorbing barrier state (calving), we derived the time for entering the absorbing state, determining the estimated time that calving occurs.

### 3.2. Patterns of Activities of Cows before Calving

We also investigated patterns in the four activities of lying, transitions from lying to standing, standing, and transitions from standing to lying. In order to do so, we raised powers to the probability matrix **P**, and look at the probabilities of diagonal elements. In other words, we researched the behavior of $p_{11}\;$, $p_{22}$, $p_{33}$ and $p_{44}$. We found that those entries in $P^{t}$ of cow ID 2 for *t* = 1, 2, 4, 6, …, 24 are represented in Table 3. As shown, the lying state, and standing state probabilities decrease when the cow approaches the calving state. However, the transition state probabilities increase. These patterns are shown in Figure 5, Figure 6 and Figure 7.

## 4. Discussion

### 4.1. Discussion on Calving Time Prediction Approaches in the Literature Surveys

Though we conducted a thorough review of the literature, we have not found any method resembling our approach, and cannot make comparisons with other methods of Markov chain analysis. However, we did find some appealing approaches in the literature, such as machine learning [15], online image analysis [16], indications of posture changes [17], and investigations of farm devices [18]. However, we feel that our approach is much easier to implement and promises comparatively favorable outcomes.

Monitoring cow behavior to predict calving events is not superficial work. In fact, no one approach could cover all aspects of monitoring cow behavior. A sizable amount of research has appeared in the literature involving the development of methods and models to predict calving time, and results have been quite promising. This section concerns a brief explanation on the topic of predicting calving time based on cow behavior monitoring, focusing on the augmented absorbing Markov chain model used to build our predictive model and analyzing the performance of our proposed method by measuring some machine learning techniques’ prediction results.

The effort of monitoring the calving process is a matter of assessing whether human assistance is required in the upcoming hours or overnight, or whether difficulties in giving birth are likely. Again, such difficulties may adversely affect production, and could even risk the life of mother and calf. Thus, an accurate and efficient method of predicting a calving event will continue to play a central role in precision dairy farming. It is no wonder that much research involving multiple disciplines has focused on predicting calving times and related research. However, we have yet to see satisfactory accomplishments in the literature.

Although a variety of unavoidable stressors continue to affect cows through calving and dry off (stopping milk production), our increased knowledge of events leading to calving should have a positive impact on milk production, as well as on cow health and overall wellbeing. The calving time prediction methods and devices can be divided into the following three categories based on:(a).Hormonal changes;(b).Clinical signs; and(c).Behavioral changes before calving.

Since the first two categories are beyond the scope of the current focus, we shall review some research that falls in the third category. The video cameras or accelerometers recording the behavior of cows can be integrated in systems using image analysis or locomotive activity to alert the dairy farmer when calving is imminent. The four comparable predictive models had been established for calving difficulty in dairy heifers and cows using four machine learning techniques: multinomial regression, decision trees, random forests, and neural networks [19]. Among many other findings, is the use of calibration evaluation techniques that have not been frequently used in agricultural or animal health applications. Apart from these models, our discussion below will extend to some other research on calving time prediction.

Some of them utilize physical measures, such as body temperature [20,21], the blood levels of progesterone, and the relaxation of pelvic ligaments [22,23]. Recently, a combination of data from sensors detecting cumulative activity, rumination activity, feeding activity, and body temperature achieved a more accurate calving time prediction system than those based exclusively on the date of insemination [24]. However, some obstacles remain to accurately predicting the starting time of calving.

Overall, systems based on behavioral analysis seem to have the most potential, because significant changes in behavior occur on the day of calving. Analyses of behavior changes normally begin several days before delivery and last until calving time. From the literature review, the most important facts and figures are as follows: searching for isolation, moving the tail, walking aimlessly, turning the head towards the abdomen, reducing rumination time, reducing the time spent lying, sniffing the ground, and frequently changing posture [25,26]. The most distinctive trends that precede calving were also noted in Santegoeds’ work [27]. In summary, these trends include the following: (1) the number of steps taken increases very slightly but significantly 10 days before calving, and more significantly over the 2 last days; (2) the time spent lying decreases slightly but significantly 10 days before calving, more significantly 3 days to 12 h before calving, and increases thereafter until after calving.

Moreover, the standing pattern is almost perfectly opposite to the lying pattern. This difference is due to an increase in time spent walking around, which interrupts periods of standing, rather than periods of lying. Walking time rises notably from two days before calving. The number of times standing up radically increases in the last6 h. Therefore, several researchers believed that close observation of cattle in the last gestation period is essential to detect the onset of calving and to reduce neonatal losses [28]. The calving time prediction is performed by using time series analysis of data on posture changes collected from video sequences recorded in the maternity barn [29,30,31]. He determined the number of transitions every hour before actual calving events using this time series analysis and could thereby predict the time of calving. Similarly, video cameras or accelerometers recording cow behavior should be integrated in systems using image analysis [32,33,34].

### 4.2. Discussion on Proposed Method

We have tested the proposed calving time prediction model using video data for pregnant dairy cows. The results are shown in Table 4. This table provides both predicted times and the actual calving times and also shows the results of using our method on data collected over a period of just 48 h. The majority of these predictions were accurate within a range of3 h. These results show great promise for practical applications in managing precision dairy farms. The results also reveal that prediction times and actual times were almost the same. These results indicate that only two days of data are needed for accurately predicting calving time. The average value of mean absolute error (MAE) for these calving time predictions is 1.101 using data collected over 72 h, and 1.229 using data collected over 48 h. We also compared our proposed method with some machine learning techniques such as K-nearest neighbors (KNN), Naïve Bayes (NB) and Support Vector Machine (SVM) by blindly testing on five cows, as shown in Table 5 and Figure 8. For the machine learning techniques, the four types of conditions comprise two postures (L, S) and two transitions (LS, SL), which are defined as four predictors. Calving and not-calving states are considered the two responses. For each cow, the calving state is defined as the response in the last 3 h before calving. During the other 69 h, the response is the not-calving state. According to the predicted calving time results of Table 4, our proposed method can accurately estimate every calving event of each cow between 69 and 73 h before the event.

From Figure 8 of the confusion matrices, the total number of observations is 360 in the testing dataset for 5 cows. In this dataset, the 2 classes are calving and not-calving, with a total of 15 calving responses and 345 not-calving responses. The best accuracy obtained using our proposed method is 100%.

## 5. Conclusions

We have developed a five-state absorbing Markov chain model to predict calving events. Although a large number of Markov chain model applications have involved research fields such as engineering, medicine and agriculture, including livestock management and animal science, we have not seen a Markov chain application that predicts calving time. Our intention was to explore and examine how Markov Models could be applied to reproduction management for dairy cows by using them to predict calving time. In this study, we only considered four types of cow behavior. Additional activities such as head movements, rumination, and raising the tail should be considered in the future. In future research, we plan to analyze some of the above-mentioned activities using the proposed Markov model. In this paper we discuss a trial-and-error method of determining parameters for the absorbing state. However, the Monte Carlo Simulation method is also attractive as a way of determining these parameters. Much remains to be done in calving time prediction research. In the future, we will combine this stochastic model with image processing techniques to detect cows, automatically recognize their behavior, and build a better model for automatically predicting calving time.

## Figures and Tables

**Figure 1 sensors-21-06490-f001:**
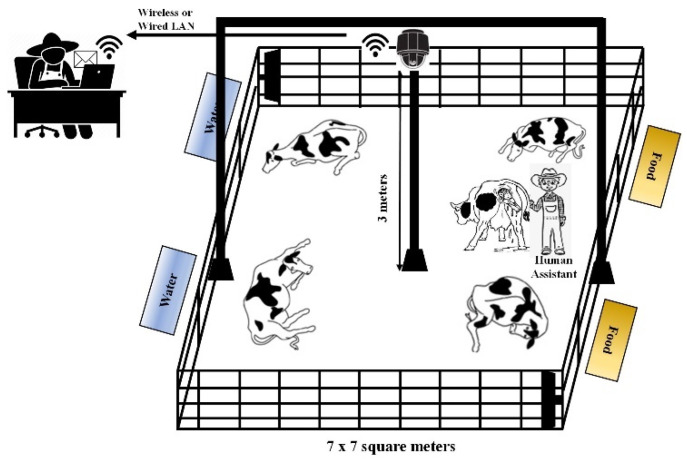
Demonstration of our research work.

**Figure 2 sensors-21-06490-f002:**
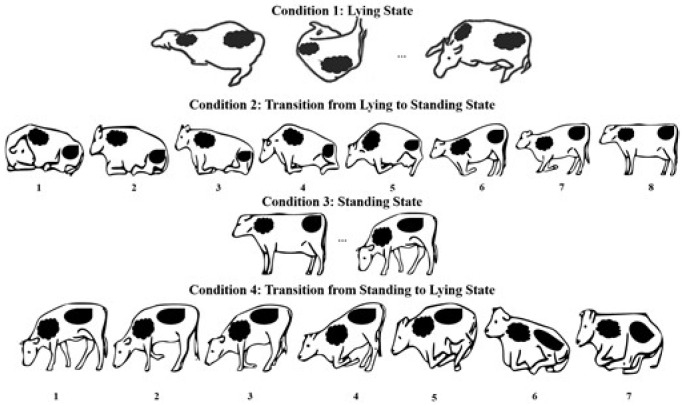
A sample of four posture conditions before calving.

**Figure 3 sensors-21-06490-f003:**
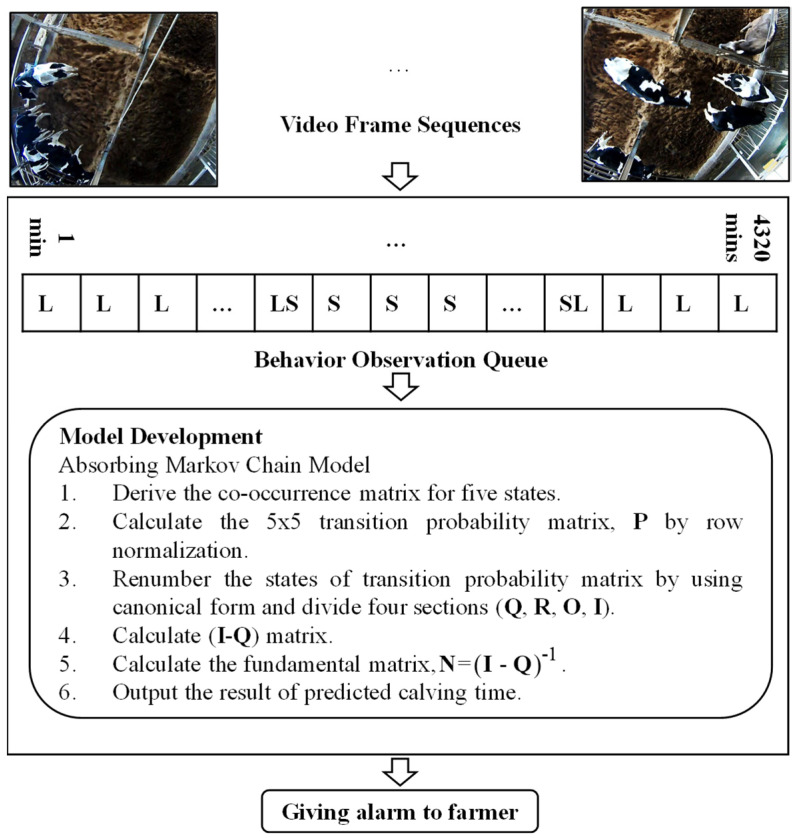
The system architecture of our proposed method.

**Figure 4 sensors-21-06490-f004:**
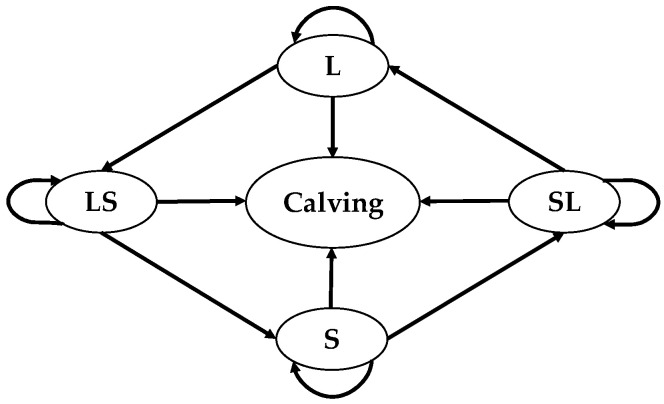
Five-state absorbing Markov Chain to assess the prediction of calving time in dairy cows.

**Figure 5 sensors-21-06490-f005:**
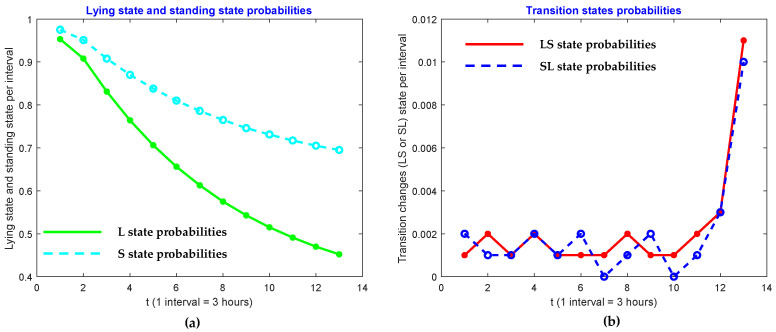
Comparison of (**a**) lying state and (**b**) transition state probabilities of ID 2.

**Figure 6 sensors-21-06490-f006:**
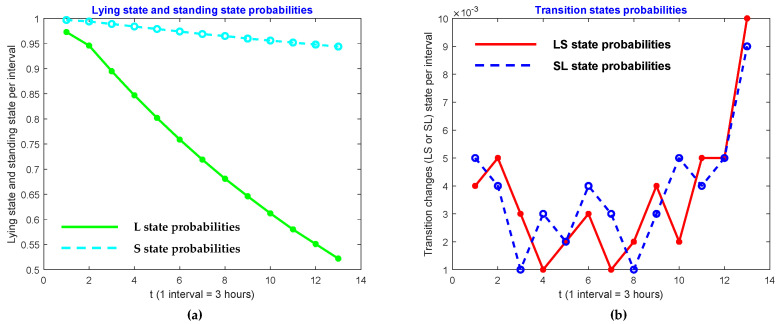
Comparison of (**a**) lying state and (**b**) transition state probabilities of ID 11.

**Figure 7 sensors-21-06490-f007:**
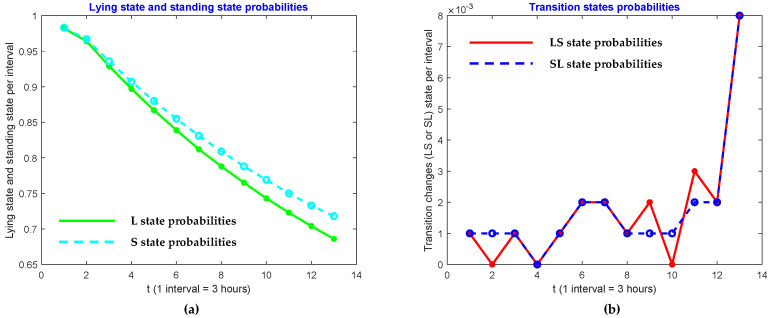
Comparison of (**a**) lying state and (**b**) transition state probabilities of ID 27.

**Figure 8 sensors-21-06490-f008:**
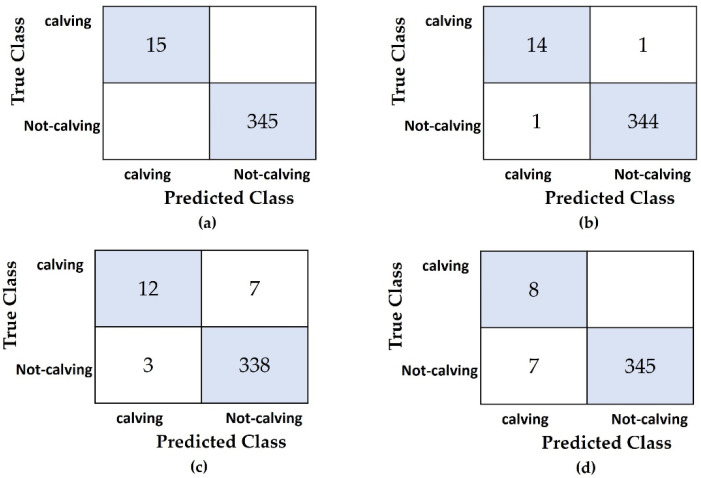
Confusion matrices of predicting calving events by using (**a**) Proposed Method, (**b**) NB, (**c**) KNN, and (**d**) SVM.

**Table 1 sensors-21-06490-t001:** Group 1: Primiparous pregnant cows data information.

Cow ID	Types of Cows	Data Collection Start Date (mm/dd/yy) and Time (h/m/s)	Calving Date (mm/dd/yy) and Time (h/m/s)
1	Black and White Holstein	11.26.2017, 17:10:00	11.29.2017, 17:10:00
3	Black and White Holstein	11.26.2017, 19:35:00	11.29.2017, 19:35:00
4	Black and White Holstein	11.30.2017, 15:10:00	12.03.2017, 15:10:00

**Table 2 sensors-21-06490-t002:** Group 2: Multiparous pregnant cows data information.

Cow ID	Types of Cows	Data Collection Start Date (mm/dd/yy) and Time (h/m/s)	Calving Date (mm/dd/yy) and Time (h/m/s)
2	Brown Swiss	11.30.2017, 21:15:00	12.03.2017, 21:15:00
5	Brown Swiss	12.09.2017, 17:15:00	12.12.2017, 17:15:00
6	Brown Swiss	12.01.2017, 10:25:00	12.04.2017, 10:25:18
7	Brown Swiss	12.03.2017, 02:40:00	12.06.2017, 02:41:22
8	Black and White Holstein	11.29.2017, 00:30:00	12.02.2017, 00:30:00
9	Black and White Holstein	12.04.2017, 10:05:00	12.07.2017, 10:06:35
10	Black and White Holstein	12.04.2017, 10:10:00	12.07.2017, 10:13:00
11	Black and White Holstein	12.04.2017, 16:05:00	12.07.2017, 16:09:40
12	Black and White Holstein	12.04.2017, 14:00:00	12.06.2017, 20:10:00
13	Black and White Holstein	12.11.2017, 06:00:00	12.14.2017, 05:58:50
14	Black and White Holstein	12.06.2017, 10:00:00	12.08.2017, 03:25:00
15	Black and White Holstein	12.12.2017, 04:50:00	12.15.2017, 04:53:09
16	Black and White Holstein	12.07.2017, 17:20:00	12.10.2017, 17:20:00
17	Black and White Holstein	12.13.2017, 21:00:00	12.16.2017, 21:03:29
18	Black and White Holstein	12.16.2017, 21:55:00	12.19.2017, 21:55:00
19	Black and White Holstein	12.14.2017, 17:15:00	12.17.2017, 17:19:00
20	Black and White Holstein	12.17.2017, 06:10:00	12.20.2017, 06:10:00
21	Black and White Holstein	12.14.2017, 16:15:00	12.17.2017, 16:17:12
22	Black and White Holstein	12.17.2017, 09:50:00	12.20.2017, 09:50:00
23	Black and White Holstein	12.15.2017, 00:50:00	12.18.2017, 01:25:21
24	Black and White Holstein	12.17.2017, 12:15:00	12.20.2017, 12:15:00
25	Black and White Holstein	11.29.2017, 00:30:00	12.02.2017, 00:30:00

**Table 3 sensors-21-06490-t003:** State probability patterns of ID2, ID11, and ID 27.

t	ID 2				ID 11				ID 27			
	*p*_11_(L)	*p*_22_(LS)	*p*_33_(S)	*p*_44_(SL)	*p*_11_(L)	*p*_22_(LS)	*p*_33_(S)	*p*_44_(SL)	*p*_11_(L)	*p*_22_(LS)	*p*_33_(S)	*p*_44_(SL)
1	0.953	0.001	0.975	0.002	0.973	0.004	0.997	0.005	0.982	0.001	0.983	0.001
2	0.908	0.002	0.951	0.001	0.946	0.005	0.994	0.004	0.964	0	0.967	0.001
4	0.831	0.001	0.908	0.001	0.895	0.003	0.989	0.001	0.929	0.001	0.936	0.001
6	0.764	0.002	0.870	0.002	0.847	0.001	0.984	0.003	0.897	0	0.907	0
8	0.706	0.001	0.838	0.001	0.802	0.002	0.979	0.002	0.867	0.001	0.880	0.001
10	0.656	0.001	0.810	0.002	0.759	0.003	0.974	0.004	0.839	0.002	0.855	0.002
12	0.613	0.001	0.786	0	0.719	0.001	0.969	0.003	0.812	0.002	0.831	0.002
14	0.575	0.002	0.765	0.001	0.681	0.002	0.965	0.001	0.788	0.001	0.809	0.001
16	0.543	0.001	0.746	0.002	0.646	0.004	0.960	0.003	0.765	0.002	0.788	0.001
18	0.515	0.001	0.731	0	0.612	0.002	0.956	0.005	0.743	0	0.769	0.001
20	0.491	0.002	0.717	0.001	0.580	0.002	0.952	0.004	0.723	0.003	0.750	0.002
22	0.470	0.003	0.705	0.003	0.551	0.005	0.948	0.005	0.704	0.002	0.733	0.002
24	0.452	0.011	0.695	0.01	0.522	0.01	0.944	0.009	0.686	0.008	0.718	0.008

**Table 4 sensors-21-06490-t004:** Experimental results of cows predicted calving time based on 72 h data and 48 h data before calving event occurs.

Cow ID	Predicted Calving Time on 72 h	Predicted Calving Time on 48 h
1	70.723	70.843
2	68.942	70.852
3	69.874	70.321
4	73.765	71.728
5	68.128	70.721
6	71.568	70.716
7	71.993	71.059
8	71.298	70.597
9	70.734	70.511
10	72.338	70.495
11	69.541	69.756
12	72.013	70.919
13	71.310	70.961
14	71.297	71.069
15	70.229	71.772
16	71.969	71.807
17	72.420	70.381
18	72.455	71.723
19	71.346	71.561
20	70.435	70.311
21	73.081	70.307
22	71.274	70.159
23	72.592	70.465
24	72.405	69.730
25	70.889	70.510

**Table 5 sensors-21-06490-t005:** Performance analysis of our proposed method by comparing with other methods.

Methods	Precision	F1 Score	Specificity	Sensitivity	Accuracy (%)
Proposed Method	1	1	1	1	100
K-nearest Neighbors (KNN)	0.890	0.846	0.811	0.811	96.100
Naïve Bayes (NB)	0.965	0.965	0.965	0.965	98.333
Support Vector Machine (SVM)	0.767	0.843	0.990	0.990	96.389

## Data Availability

The data presented in this study are available on request from the corresponding author. The data are not publicly available due to patent pending.

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
