# Peer review of "An Absorbing Markov Chain Model to Predict Dairy Cow Calving Time"

_sensors, 2021, doi:10.3390/s21196490_

Round 1

Reviewer 1 Report

Suggestions:

Include into the Introduction the machine learning techniques

Spit the Discussion section with Machine learning techniques and Discussion techniques.

Into the Machine learning section analyse the methods that the authors are proposing

Modelling of the markov chain analysis and proposing estimation techniques

Considering the structure of the paper secondary suggestions could be:

  1. In Fig 3 there is proposed method. It is not clear why this method (it is needed references) and at least it is needed one more old method for comparison
  2. In section 2.2 it is needed to propose estimation techniques and comparison with them. The section is very arbitrary
  3. In  section 2.3 a graphical presentation given the transition probabilities could be presented and analysed. How the 5 stage are connected
  4. In Fig 4,5,6 could be illustrated a model as well as estimation process for the (a) graph
  5. The analysis of machine learning techniques is unfinished. It is need to analyse methodologies, comparisons , why only these methods
  6. Major revision

Reviewer 2 Report

This paper proposes the use of an absorbing Markov chain model to predict Dairy Cow Calving Time. Statistical behaviours of pregnant cows are captured through a transition matrix between transitional states, and the expected time to calving, i.e. to enter the absorbing state, can be derived from this matrix using well known properties of absorbing Markov chains. While this approach is indeed novel for this particular application, it is not supported by meaningful results.

In section 3.1, the method for calving time expectation is only derived on one instance of cow: it has be derived for all the cows included in the study, and more global statistics at the population level should be given. In Section 3.2 are given numbers (table 2 to 4) and graphics (fig.6) that are not properly analyzed. Statistical significance of the observed changes in lying/standing transition patterns should be addressed at the level of the population, and not on 3 instances of cow only. The whole results section has to be thoroughly reconsidered in my opinion before an eventual re-submission of the paper.

Round 2

Reviewer 1 Report

ok the changes
